# IQGAP1 Regulates Actin Polymerization and Contributes to Bleomycin-Induced Lung Fibrosis

**DOI:** 10.3390/ijms25105244

**Published:** 2024-05-11

**Authors:** Tanjina Akter, Ilia Atanelishvili, Richard M. Silver, Galina S. Bogatkevich

**Affiliations:** Department of Medicine, Medical University of South Carolina, 96 Jonathan Lucas Street, Suite 912, Charleston, SC 29425, USA; takter@musc.edu (T.A.); atanelis@musc.edu (I.A.); silverr@musc.edu (R.M.S.)

**Keywords:** IQ motif containing GTPase activating protein (IQGAP1), knockout mouse, lung fibroblast, α-smooth muscle actin (SMA), scleroderma (systemic sclerosis, SSc)-associated interstitial lung disease (ILD), bleomycin

## Abstract

We previously found IQ motif containing GTPase activating protein (IQGAP1) to be consistently elevated in lung fibroblasts (LF) isolated from patients with scleroderma (systemic sclerosis, SSc)-associated interstitial lung disease (ILD) and reported that IQGAP1 contributed to SSc by regulating expression and organization of α-smooth muscle actin (SMA) in LF. The aim of this study was to compare the development of ILD in the presence and absence of IQGAP1. Pulmonary fibrosis was induced in IQGAP1 knockout (KO) and wild-type (WT) mice by a single-intratracheal instillation of bleomycin. Two and three weeks later, mice were euthanized and investigated. We observed that the IQGAP1 KO mouse was characterized by a reduced rate of actin polymerization with reduced accumulation of actin in the lung compared to the WT mouse. After exposure to bleomycin, the IQGAP1 KO mouse demonstrated decreased contractile activity of LF, reduced expression of SMA, TGFβ, and collagen, and lowered overall fibrosis scores compared to the WT mouse. The numbers of inflammatory cells and expression of pro-inflammatory cytokines in lung tissue were not significantly different between IQGAP1 KO and WT mice. We conclude that IQGAP1 plays an important role in the development of lung fibrosis induced by bleomycin, and the absence of IQGAP1 reduces the contractile activity of lung fibroblast and bleomycin-induced pulmonary fibrosis. Thus, IQGAP1 may be a potential target for novel anti-fibrotic therapies for lung fibrosis.

## 1. Introduction

IQ motif containing GTPase activating protein (IQGAP1) belongs to a family of scaffolding proteins that interact with signaling and structural molecules and regulate multiple cellular functions, including adhesion, migration, and integration of complex signaling pathways within the cell [1,2]. IQGAP1 has four isoleucine–glutamine (IQ) motifs that interact with calmodulin, protein kinases, and cell surface receptors, regulating a diverse array of target proteins [3,4].

Recently, IQGAP1 was implicated in fibrosis associated with age-related macular degeneration [5]. Additionally, IQGAP1 expression was found to be increased in nonparenchymal cells and myofibroblasts during murine liver fibrosis. Silencing IQGAP1 blocked the recruitment of bone marrow mesenchymal stromal cells and alleviated hepatic fibrogenesis [6]. Previously, we reported that IQGAP1 was elevated in lung fibroblasts isolated from scleroderma (systemic sclerosis, SSc) patients. We also found that IQGAP1 expression could be induced in normal lung fibroblasts when exposed to connective tissue growth factor (CTGF, CCN2); moreover, suppression of IQGAP1 by RNA interference decreases the migration of scleroderma and CTGF-induced normal lung fibroblasts [7]. Additionally, we demonstrated that IQGAP1 regulated the expression and organization of SMA in SSc lung fibroblasts and in thrombin-, Transforming Growth Factor (TGF)β1-, or CTGF-stimulated normal lung fibroblasts. IQGAP1 was involved in collagen gel contraction and enhanced contractility in lung fibroblasts, suggesting that IQGAP1 contributed to restrictive lung disease by regulating the expression and organization of SMA [8].

SSc is a progressive and irreversible autoimmune disease characterized by excessive collagen deposition, vascular alterations, and the production of specific autoantibodies leading to dermal and visceral organ fibrosis [9,10]. SSc-associated interstitial lung disease (ILD) occurs in 70% to 90% of SSc cases and now is a leading cause of scleroderma-related mortality [11,12]. Although the molecular mechanisms of SSc-ILD are not completely clear, the role of increased contractility of lung fibroblasts in the pathogenesis of SSc-ILD has been well recognized [13,14,15,16]. However, the factors orchestrating the contractile capacity of fibrotic lung in SSc patients and in animal models of SSc-ILD remain obscure.

While currently, no animal model of SSc-ILD completely recapitulates all pathological features of the disease, especially the progressive and irreversible condition of ILD, bleomycin-induced lung fibrosis remains informative and is widely used to study the pathogenesis of SSc-ILD [17,18]. The current study was undertaken to evaluate the role of IQGAP1 on SMA-polymerization and regulation of fibrosis in a bleomycin-induced murine model of ILD.

## 2. Results

### 2.1. Reduced Actin Polymerization in IQGAP1 Knockout Mouse

To confirm the absence of IQGAP1 protein in IQGAP1-KO mice, we employed immunoblotting analyses. IQGAP1 protein was readily detectable in lung homogenates of wild-type mice but was totally absent in lung homogenates of IQGAP1-KO mice (Figure 1A and Appendix A). We observed that IQGAP1 KO mice demonstrate reduced actin polymerization. The polymerization of actin was decreased in a time-dependent manner compared to WT mice. Additionally, growth and steady-state phases of actin polymerization were reduced as well. No significant delay was observed in the lag phase, but actin polymerization was reduced by approximately 23% during the growth phase and 32% during the steady-state phase (Figure 1B). Similar effects on actin polymerization were observed in mice with bleomycin-induced pulmonary fibrosis; the actin polymerization rate decreased by 23% during growth phase and by 29% during the steady-state phase in IQGAP1-KO mice compared to WT mice on day 21 after bleomycin administration. No significant change was observed during the lag-state phase (Figure 1C).

The cumulative rate of actin polymerization in control mice was not significantly different up to a 10 min time point during the lag phase. However, during the growth phase and steady-state phase at time points measured between 20 min and 60 min, total polymerized actin was significantly increased (*p* < 0.001) in WT mouse lung homogenate. F-actin accumulation from 31 to 40 min in the lung of the WT mouse was equal to 25,111.02 ± 2342.94 fluorescent units compared to 168,750.01 ± 1700.51 fluorescent units in the lung of IQGAP1-KO mouse (Figure 1D). A similar trend of reduction in F-actin accumulation was also observed in bleomycin-treated IQGAP-KO mice. The cumulative amount of F-actin between 31 and 40 min was equal to 22,514.80 ± 877.64 fluorescent units in the WT lung compared to 9210.21 ± 154.94 fluorescent units in the IQGAP1-KO lung (Figure 1E). To understand the effect of IQGAP1 in the regulation of actin filament formation in lung fibrosis, we compared differences in the actin polymerization rate between the saline-treated mice (∆Saline) and bleomycin-treated mice (∆Bleo). We observed a 20% reduction in the actin polymerization rate during the growth phase and a 30% reduction in the actin polymerization rate during the steady-phase state in IQGAP1-KO compared to WT mice challenged with bleomycin versus IQGAP1-KO and WT mice that received saline (Figure 1F).

### 2.2. Reduced Lung Fibrosis in IQGAP1 Knockout Mouse

Histologic evaluation of the tissue sections on day 21 after bleomycin administration showed that the lungs of saline-instilled IQGAP1-KO and WT mice appeared undistinguishable, without any significant architectural alterations. Lung histology was characterized by thin-walled alveoli, bronchioles with a smooth muscle layer, and thin connective tissue with vascular components (Figure 2A,B). Lung tissues isolated from WT mice with bleomycin-induced fibrosis demonstrated extensive peribronchial and interstitial infiltration of inflammatory cells, thickening of the alveolar walls, and multiple focal fibrotic lesions with excessive amounts of collagen and other extracellular matrix (ECM) proteins (Figure 2C). Bleomycin-induced pulmonary fibrosis was less severe in IQGAP1-KO mice (Figure 2D). The overall fibrotic scores were assessed quantitatively by the Ashcroft system [19]. The Ashcroft score in bleomycin-challenged IQGAP1-KO mice was significantly lower than in the bleomycin-challenged WT mice (*p* < 0.05, Figure 2Q).

To evaluate the effect of IQGAP1 on collagen accumulation in a bleomycin-induced pulmonary fibrosis mouse model, we quantified total soluble collagen content in the lung tissue using the Sircol collagen assay. Bleomycin increased soluble collagen content 3.7-fold in WT mice but only 2.4-fold in IQGAP1-KO mice (*p* < 0.05, Figure 2R).

The fluorescent immunohistochemistry of lung tissue with Transforming Growth Factor (TGF)β antibodies showed that control lung tissue had less than 20% of TGFβ-positive cells in both IQGAP1 KO and WT mice. Bleomycin caused an increase in TGFβ positive cells in WT mice to 40.8 ± 1.7%. A significantly smaller number of TGFβ-positive cells (23.2 ± 5.9%, *p* < 0.05) was observed in the IQGAP1 KO mice group (Figure 2E–H,S).

SMA histochemistry showed a very low (less than 5%) number of cells expressing SMA in the control lung tissue. We observed a 29.6-fold increase in the SMA-expressed cells in bleomycin-treated WT mice but only a 9.2-fold increase in the bleomycin-instilled IQGAP1 KO mice (Figure 2I–L,T). Statistical analysis showed that the number of SMA-positive cells in bleomycin-challenged WT mice was significantly higher (*p* < 0.05) than in bleomycin-challenged IQGAP1 KO mice (47.5 ± 10.7% and 28.6 ± 6.9%, respectively).

To compare the expression of IQGAP1 between control and bleomycin-challenged mice, we performed immunohistochemistry with anti-IQGAP1 antibody. We observed significantly higher expression of IQGAP1 in lung tissue isolated from bleomycin-challenged WT mice compared to lung tissue isolated from WT control mice (Figure 2M,O,U). As expected, no IQGAP1-positive cells were present in lung tissue obtained from IQGAP1 KO mice (Figure 2N,P).

Next, we isolated total lung RNA and measured mRNA levels of collagen type I, SMA, and TGFβ by qRT-PCR in all groups sacrificed 3 weeks after bleomycin administration. We found that deletion of IQGAP1 results in lower transcriptional levels of bleomycin-induced collagen type I, SMA, and TGFβ in comparison to WT mice (Figure 3).

### 2.3. Studies of the Pro-Inflammatory Components of Bleomycin-Induced Lung Injury

To determine whether IQGAP1 expression is associated with bleomycin-induced pro-inflammatory signals in the lung, we performed broncho-alveolar lavage in mice sacrificed 2 weeks after bleomycin administration. We observed that protein content and cell counts in broncho-alveolar lavage (BAL) fluid on day 14 were markedly higher in the bleomycin-treated mice compared to saline-treated mice (Figure 4A). We found there to be no significant difference between IQGAP1 KO and WT groups of mice. The transcriptional levels of bleomycin-induced pro-inflammatory cytokines tumor necrosis factor (TNF)-*α* and interleukin (IL)-6 were also not affected by the deletion of IQGAP1 (Figure 4B).

### 2.4. Contractile Activity of Lung Fibroblasts Isolated from IQGAP1 KO Mouse

Lung fibroblasts isolated from IQGAP1 KO and WT groups of mice were cultured on gelatin-coated slides and stained with SMA antibodies. We observed that lung fibroblasts isolated from bleomycin-treated WT mice expressed a high amount of SMA, whereas lung fibroblasts isolated from saline-treated WT mice expressed visibly less SMA (Figure 5A). Lung fibroblasts isolated from either bleomycin- or saline-treated IQGAP1 KO mice were characterized by a low amount of SMA.

To study the contractile activity of IQGAP1 KO lung fibroblasts, we performed collagen gel contraction assays. We previously reported that cultured lung fibroblasts isolated from bleomycin-treated mice are characterized by higher contractile activity compared to control lung fibroblasts [20]. We observed that lung fibroblasts from bleomycin-treated WT mice contracted collagen gels from 15 mm in diameter to less than 8 mm in diameter within 24 h (8.2 ± 1.8 mm contraction). In contrast, contraction from bleomycin-treated IQGAP1 KO mice reached only 4.8 ± 1.2 mm (Figure 5B). Collagen gel contraction of lung fibroblasts from saline-treated mice was not affected by IQGAP1.

## 3. Discussion

Knockout mouse models have proven to be highly useful tools in helping to understand the causality of genes on cellular levels and in the pathogenesis of diseases. In this study, we employed IQGAP1 KO mice developed on C57BL/6 and 129J1 backgrounds to comprehend the functional roles of IQGAP1 in interstitial lung disease (ILD). Our study shows that lung tissue from IQGAP1-KO and WT mice appeared undistinguishable, without any significant architectural alterations. However, at the molecular level, the lack of IQGAP1 significantly affected actin polymerization. The role of IQGAP1 in actin polymerization has been extensively investigated, and several different mechanisms have been proposed [21,22,23]. One mechanism involves stimulation of actin assembly through the myosin light chain (MLC) pathway [21]; another is based on promoting the polymerization and branching of actin through the neuronal Wiskott–Aldrich syndrome protein–actin-related protein 2/3 (N-WASP-Arp2/3) pathway [22]. A third mechanism is based on the direct interaction of IQGAP1′s calponin homology (CH) domain with polymerized F-actin. It was reported that the CH domain of IQGAP1 facilitates actin polymerization by cross-linking filaments into interconnected bundles [23].

To induce ILD in IQGAP1 KO mice, we utilized bleomycin, a classical antineoplastic drug initially isolated from a strain of actinobacteria, *Streptomyces verticillus* [24]. ILD is a well-known side effect of bleomycin because of very low levels of the bleomycin-inactivating enzyme called bleomycin hydrolase in the lungs [25]. Intratracheal administration of bleomycin causes pulmonary injury, inflammation, and subsequent fibrosis [26]. The scaffold protein IQGAP1 is implicated in several pathways commonly associated with fibrosis, including adhesion, migration, and contractile activity of lung myofibroblasts [7,8,27]. However, direct proof of IQGAP1′s involvement in pulmonary fibrosis in vivo has been lacking. Moreover, findings on IQGAP1 expression in pulmonary fibrosis are somewhat contradictory. Kulkarni et al. reported that expression of IQGAP1 is increased in lung tissue from mice with bleomycin-induced pulmonary fibrosis [28], while Zong et al. showed that expression of IQGAP1 is reduced in lung fibroblasts obtained from mice with bleomycin-induced pulmonary fibrosis [29]. We previously reported that lung fibroblasts isolated from patients with scleroderma-associated pulmonary fibrosis are characterized by higher expression of IQGAP1 compared to control lung fibroblasts [7]. Our current study demonstrates significantly higher expression of IQGAP1 in lung tissue from mice with bleomycin-induced pulmonary fibrosis. We show that IQGAP1 KO mice are less susceptible to bleomycin-induced ILD compared with WT mice. Lung tissues from bleomycin-challenged IQGAP1-KO mice demonstrated decreased expression of SMA and TGF-β, less histologic alteration, and reduced overall fibrosis compared to the WT mice.

The role of inflammation in the pathogenesis of progressive pulmonary fibrosis remains a controversial issue. Bleomycin causes a severe acute inflammatory response followed by chronic inflammation and fibrosis [26]. It has been shown that the degree of inflammation in bleomycin-induced pulmonary injury is associated with the intensity of fibrosis [26]. The role of IQGAP1 in inflammation is unclear and might be tissue-dependent. IQGAP1 was shown to play a pro-inflammatory function in angiogenesis- and inflammation-dependent ischemic cardiovascular diseases [30]. Inactivating IQGAP1 normalized Rac1 GTP-loading and reduced inflammation and arthritis in GTPase-I-deficient mice, preventing statins from increasing Rac1 GTP-loading and cytokine production in macrophages [31]. Deficiency of IQGAP1 significantly attenuated foreign body response by deactivating pro-inflammatory and profibrotic pathways [32]. In contrast, the experimentally induced brain and spinal cord inflammation was more severe in IQGAP1-KO mice than in WT littermates [33]. In our study, we could not identify any significant effects of IQGAP1 on bleomycin-induced pro-inflammatory signaling, including cell counts and protein levels in BAL fluid or levels of pro-inflammatory cytokines TNF-α and IL-6. These data may suggest that resistance of IQGAP1-KO mice to bleomycin-induced pulmonary fibrosis involves anti-fibrotic but not anti-inflammatory mechanisms.

Fibroblasts in active fibrotic lesions differentiate into ultrastructural and metabolically distinctive myofibroblasts identified as key mesenchymal cells responsible for tissue remodeling and fibrosis [34]. The actin cytoskeleton of myofibroblasts consists of SMA, β-actin, and γ-actin isoforms, present in cells in either a monomeric state (G-actin) or a polymeric state (F-actin). G-actin, also called cytosolic actin, localizes mainly in the cell cytosol, while bundles of polymerized F-actin form the actin stress fiber network [35,36]. Our previous data demonstrate that in SSc-ILD myofibroblasts, IQGAP1 shuttles monomeric SMA, facilitating its assembly through a mechanism distinct from β- and γ-actin polymerization [8], which suggests that shuttling monomeric SMA results in the increased rate of SMA polymerization and organization in lung fibroblasts.

Here, we show that IQGAP1 KO mice demonstrate a reduced rate of actin polymerization and reduced accumulation of actin in the lungs, followed by reduced expression of SMA following bleomycin administration. Lung fibroblasts isolated from bleomycin-administrated WT mice exhibited higher expression of SMA and more profound contractile activity, suggesting that IQGAP1 is an important regulator of SMA expression and contractile forces in bleomycin-induced pulmonary fibrosis.

In conclusion, our results identify IQGAP1 as a scaffolding protein contributing to bleomycin-induced pulmonary fibrosis and strongly suggest that IQGAP1 participates in SSc-ILD pathophysiology by increasing contractile forces and lung stiffness. Thus, IQGAP1 may be a novel target for the development of much-needed anti-fibrotic therapy for SSc-ILD as well as for other diseases characterized by lung fibrosis.

## 4. Materials and Methods

### 4.1. Generation of IQGAP1-KO Mice

IQGAP1 KO mice were developed on C57BL/6 and 129J1 backgrounds by Dr. A. Bernards at the Massachusetts General Hospital Cancer Center. The murine *Iqgap1* gene was inactivated by the deletion of 233-bp murine exon containing the NLLYYRYMNPAIVAP RasGAP signature motif [35]. IQGAP1 KO mice and matching wild-type (WT) mice were obtained from Dr. Wadie Bahou (Stony Brook University, New York, NY, USA) and were bred in the Division of Laboratory Animal Resources of the Medical University of South Carolina (MUSC) in accordance with guidelines of the Institutional Animal Care and Use Committee (IACUC) of the MUSC protocol AR#2921. The absence of IQGAP1 in the lungs of the IQGAP1 KO mouse was confirmed by immunoblotting with anti-IQGAP1 antibody (Santa Cruz Biotechnology, Santa Cruz, CA, USA) as detailed in Appendix A.

### 4.2. Bleomycin-Induced Mouse Model of ILD

All mouse-related experimental procedures were performed according to the guidelines of the IACUC MUSC AR#2921. Lung injury was induced in 8-week-old IQGAP1 KO mice and age- and sex-matched WT mice on C57BL/6 and 129J1 backgrounds by intratracheal instillation of bleomycin (Teva Pharmaceuticals, Parsippany, NJ, USA) at a concentration of 2 U/kg in 50 µL of saline. Control mice received the same volume of saline. The mice were grouped as follows: (1) WT control group; (2) KO control group; (3) WT bleomycin group; and (4) KO bleomycin group. All mice were maintained in animal quarters specially designated for pathogen-free mice and were provided with food and water ad libitum. Mice were sacrificed two and three weeks after bleomycin instillation; lung tissue and BAL fluid were investigated.

### 4.3. Lung Fixation and Histological Examinations

Sacrificed mice were subjected to midline thoracotomy. The trachea was cannulated, and the lungs were fixed by instillation of buffered formalin (4%) for 24 h, followed by perfusion with 70% ethanol for another 24 h before routine processing and paraffin embedding. To evaluate the stages of lung fibrosis, multiple sections from each lung were stained with hematoxylin and eosin (H&E staining) or with trichrome staining for collagen and other ECM proteins. Fibrosis quantification was performed using 0 (normal) to 8 (total fibrosis) scores to calculate an Ashcroft score, as previously described [20]. Morphological changes, such as the thickness of alveolar septae, accumulation of vascular components and connective tissue, and the infiltration by inflammatory cells, were also analyzed. For histological evaluation, each specimen was divided into 10 non-overlapping fields and scored independently. To avoid bias, all histological specimens were evaluated by three individuals in a blinded fashion. The mean value from the individual score is presented as the fibrotic score. TGF-β and SMA immunohistochemistry was performed using rabbit polyclonal anti-TGF-β1 antibody from Santa Cruz Biotechnology (Santa Cruz, CA, USA) and anti-α-SMA rabbit polyclonal antibody from Abcam (Cambridge, MA, USA). IQGAP1 immunohistochemistry was performed using mouse monoclonal anti-IQGAP1 antibody from Santa Cruz Biotechnology (Santa Cruz, CA, USA).

### 4.4. Collagen Assay

Collagen assay was performed using the Sircol collagen assay method from Accurate Chemical and Scientific Corp. (Westbury, NY, USA) in accordance with the manufacturer’s instructions.

### 4.5. Actin Polymerization Assay

The experiments were performed using the Actin polymerization Biochem kit (Cytoskeleton, Denver, CO, USA), in which the rate of pyrene-labeled G-actin conversion into F-actin was monitored by fluorescent signal. For sample preparation, the whole mouse lung was homogenized in 10 mL of extraction buffer (20 mM NaCl, 20 mM Tris-HCl, pH 7.5, and the protease inhibitor cocktail) on dry ice using a mechanical homogenizer, followed by a short pulse sonication and centrifugation at 100,000× *g* for 1 h. The supernatant was collected and ultrafiltrated to a protein concentration of 10 mg/mL using an Amicon ultra-0.5 centrifugal filter. Pyrene fluorescence signals were monitored in clear flat-bottom black-walled 96-well plates using the FLIPR^tetra^ system (Fluorometric Imaging Plate Reader, Molecular Devices, Sunnyvale, CA, USA) with fluorescence excitation LED 360–380 nm and an emission filter of 400–460 nm. Pyrene-labeled rabbit skeletal muscle G-actin was used as a polymerization positive control. A total of three independent experiments were performed with 4 mice per experiment.

### 4.6. RNA Isolation and RT-PCR Analysis

Total lung tissue RNAs were isolated from all groups of mice using the RNA Isolation Kit from Qiagen (Valencia, CA, USA) according to the manufacturer’s recommendations. RNA purity and amount isolated were determined by spectrophotometric analysis. Reverse transcription was performed with the SuperScript II First-Strand Synthesis Kit from Invitrogen (Carlsbad, CA, USA), and RT-PCR was performed with SYBR Green PCR Master Mix Kit from Bio-Rad (Hercules, CA, USA). PCR primers, synthesized by Eurofins Genomics (Louisville, KY, USA), are presented in Appendix A. RT-PCR was performed on a Bio-Rad MyIQ single-color real-time PCR detection system under the following conditions: 95 °C for 3 min, followed by 35 cycles at 95 °C for 30 s and 60 °C for 1 min. Relative differential expression of genes was calculated using the method described by Pfaffl [36], with Glyceraldehyde-3-phosphate dehydrogenase (GAPDH) serving as a housekeeping gene. Relative expression of mRNAs in saline-treated mice was normalized as 100% or “one” for fold change calculations.

### 4.7. Lung Fibroblast Isolation and Culture

Lung tissue was diced (0.5 × 0.5 mm pieces) and cultured on 6-well plates covered with 1% gelatin in Dulbecco’s modified Eagle’s medium (DMEM; Gibco, Grand Island, NY, USA) supplemented with 10% fetal bovine serum, 2 mM l-glutamine, 50 μg/mL of gentamicin sulfate, and 5 μg/mL of amphotericin B at 37 °C in 10% CO_2_. The medium was changed every three days to remove dead and non-attached cells until fibroblasts reached confluence. Monolayer cultures were maintained in the same medium. Lung fibroblasts were used between the second and the fourth passages in all experiments. The viability of all cells used in this study was confirmed by trypan blue cell counting assay in which cell suspension was mixed with Trypan blue solution at a ratio of 1:1, and viable cells were counted under light microscopy using a hemacytometer.

### 4.8. Collagen Gel Contraction Assay

Collagen lattices were prepared with type I collagen from rat tail tendon adjusted to a final value of 2.5 mg/mL with 0.01% acetic acid. Lung fibroblasts at a concentration of 2.5 × 10^5^ cells/mL were suspended in collagen (1.25 mg/mL of collagen) and aliquoted into 24 well plates (300 µL/well). Collagen lattices were polymerized for 45 min in a humidified 10% CO_2_ atmosphere at 37 °C, followed by incubation with DMEM containing 10% FBS for 4 h. To determine the degree of collagen gel contraction, pictures were taken after 24 h with a digital camera. Measurement of the diameter of each gel in millimeters (mm) was recorded as the average values of the major and minor axes. Calculation of gel contraction was presented as the difference between the diameters of wells and contracted gels.

### 4.9. Immunofluorescent Studies in Isolated Lung Fibroblasts

Mouse lung fibroblasts were cultured to sub-confluence on glass slides covered with 1% gelatin in DMEM containing 10% FBS, after which cells were washed with cold PBS, fixed in methanol at −20 °C for 4 min, and washed with cold PBS twice, followed by incubation with anti-α-SMA rabbit polyclonal antibody from Abcam (Cambridge, MA, USA) (1:500) for 1 h at room temperature. Cells were washed three times with cold PBS, incubated with Alexa Fluor 488 anti-rabbit IgG (1:200) and DAPI (1:10,000) for 1 h at room temperature, then washed with cold PBS, air-dried, covered, and sealed. Images were acquired with an Olympus IX71 fluorescence microscope (Olympus, Tokyo, Japan) equipped with objective x60/1.42 and Olympus Slidebook 4.1 software.

### 4.10. Statistical Analysis

Statistical analyses were performed using two-way analysis of variance models followed by post hoc testing or nonparametric test as appropriate with KaleidaGraph 4.0 (Synergy Software, Reading, PA, USA). Actin polymerization was analyzed by independent sample *t*-test on IBM SPSS statistical software (version 24 for Windows; SPSS, Chicago, IL, USA). All results were considered significant if *p* < 0.05.

## Figures and Tables

**Figure 1 ijms-25-05244-f001:**
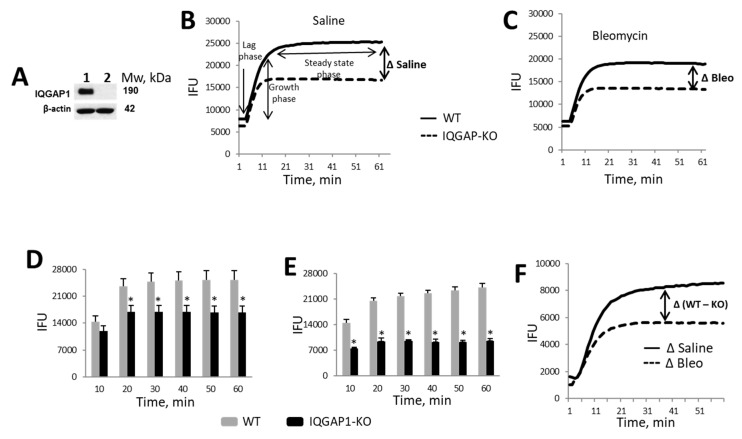
Reduced actin polymerization in IQGAP1 knockout mouse. (**A**) IQGAP1 is expressed in lung homogenates of wild-type (WT) mice (lane 1) but not in IQGAP1-KO mice (lane 2). (**B**–**F**) actin polymerization rate in WT and IQGAP1-KO mice presented in integrated fluorescence units (IFU). (**B**,**C**) actin polymerization rate between lung homogenates of WT and IQGAP1-KO mice were compared after intratracheal saline (**B**) and bleomycin (**C**) instillation. The actin polymerization rate difference between WT and IQGAP1-KO is presented as ∆. (**D**,**E**) cumulative amount of F-actin in lung of IQGAP1-KO and WT control (**D**) or bleomycin-instilled (**E**) mice at different time points are presented. The values are mean and SE from twelve mice/three independent experiments; asterisks represent statistically significant differences in total amount of F-actin formation between IQGAP1-KO and WT mice (*p* < 0.001). (**F**) difference in the rate of actin polymerization between WT and IQGAP1-KO compared after intratracheal instillation of saline (Δ1_Saline) and bleomycin (Δ2_Bleo).

**Figure 2 ijms-25-05244-f002:**
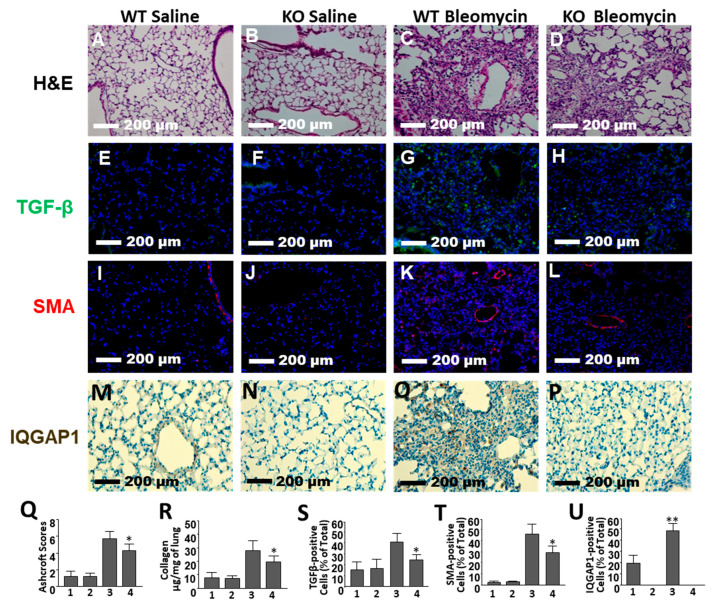
Effect of IQGAP1 knockout on bleomycin-induced pulmonary fibrosis. Representative lung tissue sections from wild type control mice (WT saline, images (**A**,**E**,**I**,**M**)), IQGAP1-KO saline (images (**B**,**F**,**J**,**N**), WT bleomycin (images (**C**,**G**,**K**,**Q**)), and IQGAP1-KO bleomycin (images (**D**,**H**,**L**,**P**)), stained with Hematoxylin and Eosin (H&E, images (**A**–**D**)), anti-TGFβ1 antibody (images (**E**–**H**)), anti-SMA antibody (images (**I**–**L**)), and anti-IQGAP1 antibody (images (**M**–**P**)) are presented. (**Q**) quantitative evaluation of fibrotic changes (Ashcroft scores), n = 32 (8 mice per group). (**R**) collagen lung content determined by Sircol collagen assay. (**S**) TGFβ1 expression. (**T**) SMA expression. (**U**) IQGAP1 expression. Bars in (**Q**–**U**) are labeled as follows: 1—WT saline; 2—IQGAP1-KO saline; 3—WT bleomycin; 4—IQGAP1-KO bleomycin. Number of mice for (**R**,**S**,**T**) is 24 (6 mice per group); number of mice for (**U**) is 12 (3 mice per group). Values in (**Q**–**U**) are the mean and SD. * Statistically significant differences between IQGAP1-KO bleomycin versus WT bleomycin mice (*p* < 0.05). ** Statistically significant differences between WT bleomycin versus WT saline mice (*p* < 0.05).

**Figure 3 ijms-25-05244-f003:**
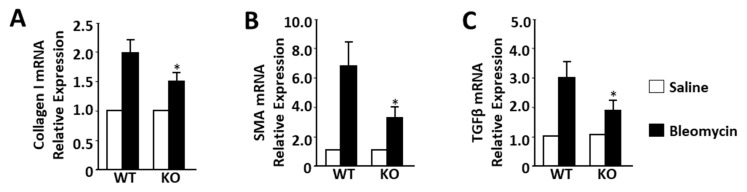
Effect of IQGAP1 depletion on collagen type I mRNA (**A**), SMA mRNA (**B**), and TGFβ mRNA (**C**) expression in lung from bleomycin challenged mouse. Data are presented as mean ± SD; n = 24 (six mice per group). Relative expression of mRNA in saline-treated mice was normalized as 100% or “one” for fold change calculations. The asterisk represents statistically significant differences (*p* < 0.05) between IQGAP1-KO bleomycin and WT bleomycin mice.

**Figure 4 ijms-25-05244-f004:**
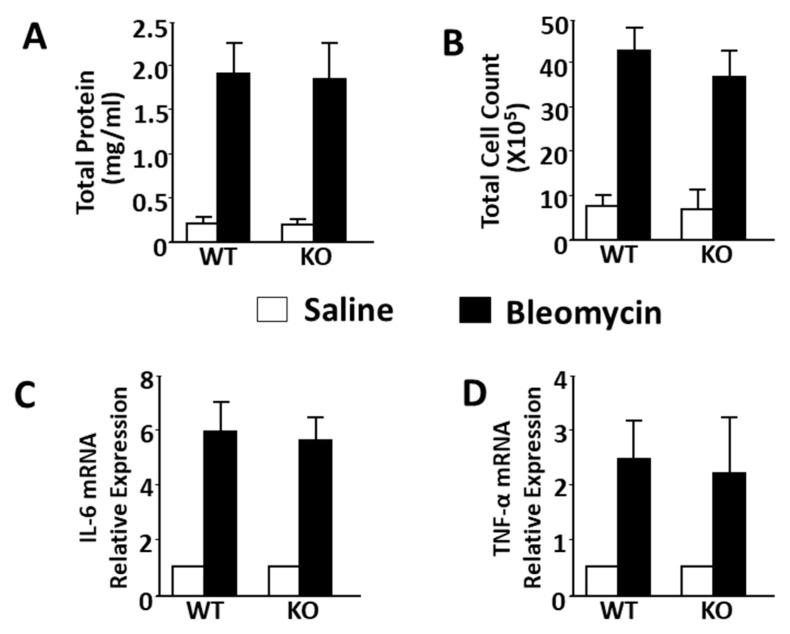
Effect of IQGAP1 depletion on inflammatory reaction induced by bleomycin in mice. (**A**,**B**), effect of IQGAP1 on bleomycin-induced protein (**A**) and inflammatory cell infiltration (**B**) in BALF. (**C**,**D**), effect of IQGAP1 on bleomycin-induced IL-6 mRNA and TNF-α mRNA expression in mouse lung. Data are presented as mean ± SD. Relative expression of IL-6 mRNA and TNF-α mRNA in saline-treated mice was normalized as 100% or “one” for fold change calculations. n = 24 (6 mice per group).

**Figure 5 ijms-25-05244-f005:**
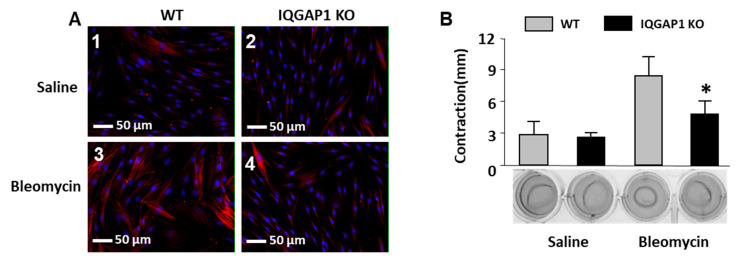
Effect of IQGAP1 on SMA expression (**A**) and collagen gel contraction (**B**) in mouse lung fibroblasts. 1—WT saline; 2—IQGAP1 KO saline; 3—WT bleomycin; 4—IQGAP1 KO bleomycin, n = 24 (6 mice per group). SMA expression in (**A**) is shown as red fluorescent staining. Nuclei are stained blue by 4′,6-diamidino-2-phenylindole (DAPI). Values in (**B**) are the mean ± SD. * = *p* < 0.05 IQGAP1-KO bleomycin versus WT bleomycin mice.

## Data Availability

The datasets used and/or analyzed during the current study period are available from the corresponding author upon request.

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
