# Peer review of "IQGAP1 Regulates Actin Polymerization and Contributes to Bleomycin-Induced Lung Fibrosis"

_ijms, 2024, doi:10.3390/ijms25105244_

Round 1

Reviewer 1 Report

Comments and Suggestions for Authors

This study presents compelling evidence regarding the role of IQ motif containing GTPase activating protein (IQGAP1) in the development of pulmonary fibrosis. By comparing IQGAP1 knockout (KO) mice with wild-type (WT) mice following bleomycin-induced pulmonary fibrosis, the researchers demonstrate a significant reduction in fibrotic response and contractile activity of lung fibroblasts in the absence of IQGAP1.

The findings suggest that IQGAP1 is intricately involved in regulating the expression and organization of α-smooth muscle actin (SMA) in lung fibroblasts, thereby contributing to the progression of fibrosis. Notably, the study highlights the potential of IQGAP1 as a promising target for novel anti-fibrotic therapies aimed at mitigating lung fibrosis.

While the study provides evidence of the involvement of IQGAP1 in regulating α-smooth muscle actin (SMA) expression and fibrosis development, the exact molecular mechanisms underlying these processes remain incompletely understood. In the future, mechanistic studies are needed to elucidate the precise pathways through which IQGAP1 influences fibrotic processes.

Major Point:

The authors previously identified elevated levels of IQ motif-containing GTPase activating protein (IQGAP1) consistently in lung fibroblasts (LF) isolated from patients with scleroderma (systemic sclerosis, SSc)-associated interstitial lung disease (ILD). In this study, a mouse animal model was employed. It is needed to confirm the dysregulation of Iqgap1 in the bleomycin-treated mouse lung. This confirmation could be achieved by using the lung tissues through Western blotting, Real-time PCR, or immunostaining. Alternatively, assessing the expression of Iqgap1 in lung fibroblasts isolated from bleomycin-treated mice through Western blotting or Real-time PCR could be considered.

Minor Points:

1.      In Figure 1D and E, it is noted that the data come from three independent experiments. However, it is unclear how many mice were used in each experiment.

2.      In Figure 3 and Figure 4C and D, there are no error bars on the saline group. Further clarification is needed regarding this omission, whether it is due to the calculation of fold change necessitated by different batches of animals or other reasons.

3.      In Figure 3, the number of samples (N number) needs to be provided.

4.      In the Methods section regarding the Actin polymerization assay, additional information is required concerning the number of animals used in each experiment.

5.      In the Methods section, regarding the Actin polymerization assay, clarification is needed regarding "10 mg/ml." Is this referring to the protein concentration?

Author Response

Reviewer 1:

Major Point:

The authors previously identified elevated levels of IQ motif-containing GTPase activating protein (IQGAP1) consistently in lung fibroblasts (LF) isolated from patients with scleroderma (systemic sclerosis, SSc)-associated interstitial lung disease (ILD). In this study, a mouse animal model was employed. It is needed to confirm the dysregulation of Iqgap1 in the bleomycin-treated mouse lung. This confirmation could be achieved by using the lung tissues through Western blotting, Real-time PCR, or immunostaining. Alternatively, assessing the expression of Iqgap1 in lung fibroblasts isolated from bleomycin-treated mice through Western blotting or Real-time PCR could be considered.

Response: Absence of IQGAP1 in lung from IQGAP1 KO mouse was confirmed by immunoblotting with anti-IQGAP1 antibody (Santa Cruz Biotechnology, Santa Cruz, CA) as detailed in Supplementary Materials and showed in main manuscript Figure 1A and in Supplementary Materials Figure 1.

Minor Points:

  1. In Figure 1D and E, it is noted that the data come from three independent experiments. However, it is unclear how many mice were used in each experiment.

Response: The data presented in Figure 1 D and E were obtained from three independent experiments with four mice being used in each experiment (12 mice total). The corresponding statement was added to Figure 1 legend.

  1. In Figure 3 and Figure 4C and D, there are no error bars on the saline group. Further clarification is needed regarding this omission, whether it is due to the calculation of fold change necessitated by different batches of animals or other reasons.

Response: Relative expression of mRNA in saline-treated mice was normalized as 100% or “one” for fold change calculations. The corresponding statement was added to the legends of both figures.

  1. In Figure 3, the number of samples (N number) needs to be provided.

Response: n = 24 (six mice per group) has been provided.

4 and 5.      In the Methods section regarding the Actin polymerization assay, additional information is required concerning the number of animals used in each experiment; clarification is needed regarding "10 mg/ml." Is this referring to the protein concentration?

Response: Additional information in the Methods section (Actin Polymerization Assay) has been provided. It was clarified that a total of three independent experiments were performed with 4 mice per each experiment and that 10 mg/ml is referring to the protein concentration.

Reviewer 2 Report

Comments and Suggestions for Authors

1. The author may add a small description of the IQ motif.

2. The author should write IQ motif containing GTPase activating protein 1 (IQGAP1)

3 . The author should give details of the abbreviations of TGFβ. 

4. In the introduction the author should cite 1 and 2 as per format.

5. In the introduction, 2nd paragraph citations no. 4 and 5 came twice. The author should cite at the end of the statement from the cited article. 

6. In Figure 2 (M-P), the author didn't describe labels 1,2,3,& 4 in the bar diagram.

7. In Figure 5 B, the author should use a larger image of the collagen gel contraction. 

8. Overall discussion is well-written however, the author didn't discuss the obtained result as appeared in the result section.

9. In the materials and methods section, the author may provide an institutional ethical clearance number.

10. In 4.1 Generation of IQGAP1..., It is not clear that used mice developed by the author as described in the section or borrowed and bred in their facility. 

11. What stands for BAL fluid, should mentioned?

12. In lung fixation and histological examinations, it appeared that the author fixed lungs in 2% formalin for 24 hours and then perfused with 70% ethanol for 24 hours. The author should justify the methodology and conditions of perfusion. 

13. In 4.5, Actin polymerization assay, the whole mouse lung was homogenized in extraction buffer, Author should provide the volume of extraction buffer, in which the mouse lung was homogenized. 

14. In RNA isolation and RT-PCR analysis, primers designed by the author or used from published articles. The author should provide details.

15. Each primer has a specific melting point, the author should provide melting points of the primer along with primers in a table.

16. In lung fibroblast isolation and culture, the viability of cells assed by ApoSENSORTM cell viability assay or trypan blue cell counting assay. Which method was used by the author in this study? 

17. In collagen gel contraction assay, line no. 334, type 1 collagen from rat tail......., it may be mouse tail.

18. In statistical analysis, which ANOVA was used? 

Author Response

Reviewer 2:

1 and 2. The author may add a small description of the IQ motif. The author should write IQ motif containing GTPase activating protein 1 (IQGAP1).

Response: Such information was added to the Introduction. See “IQGAP1 has four isoleucine-glutamine (IQ) motifs that interacts with calmodulin, protein kinases and cell surface receptors regulating a diverse array of target proteins.”

3 . The author should give details of the abbreviations of TGFβ. 

Response: Full form of TGFβ as Transforming Growth Factor β has been clarified.

  1. In the introduction the author should cite 1 and 2 as per format.

Response: Corrected.

  1. In the introduction, 2nd paragraph citations no. 4 and 5 came twice. The author should cite at the end of the statement from the cited article. 

Response: Corrected.

  1. In Figure 2 (M-P), the author didn't describe labels 1,2,3,& 4 in the bar diagram.

Response: Figure legends were clarified to show description of labels 1,2,3&4.

  1. In Figure 5 B, the author should use a larger image of the collagen gel contraction. 

Response: As per reviewer’s suggestion, a larger image of the collagen gel contraction was used.  

  1. Overall discussion is well-written however, the author didn't discuss the obtained result as appeared in the result section.

Response: The discussion was revised to further interpretate the presented results.

  1. In the materials and methods section, the author may provide an institutional ethical clearance number.

Response: The institutional ethical clearance number has been added to the material and methods section.

  1. In 4.1 Generation of IQGAP1..., It is not clear that used mice developed by the author as described in the section or borrowed and bred in their facility. 

Response: “Generation of IQGAP1 KO” section of Materials and Methods was revised to clarify that IQGAP1 KO mice and wild type control mice were obtained from Dr. Wadie Bahou (Stony Brook University) and bred at the Medical University of South Carolina (MUSC) in accordance with guidelines of the Institutional Animal Care and Use Committee (IACUC) of the MUSC protocol AR#2921.

  1. What stands for BAL fluid, should mentioned.

Response: Full form of BAL as Bronchoalveolar Lavage has been clarified.

  1. In lung fixation and histological examinations, it appeared that the author fixed lungs in 2% formalin for 24 hours and then perfused with 70% ethanol for 24 hours. The author should justify the methodology and conditions of perfusion. 

Response: Lung fixation and histological examinations has been clarified that 4% formalin for 24 hours followed with 70% ethanol for 24 hours are being used in our laboratory as per established protocol.

  1. In 4.5, Actin polymerization assay, the whole mouse lung was homogenized in extraction buffer, Author should provide the volume of extraction buffer, in which the mouse lung was homogenized. 

Response: The volume of extraction buffer (10 ml) has been added to the Methods.

  1. In RNA isolation and RT-PCR analysis, primers designed by the author or used from published articles. The author should provide details.

Response: All primers were designed in the laboratory using NCBI tool (https://blast.ncbi.nlm.nih.gov/Blast.cgi). 

  1. Each primer has a specific melting point, the author should provide melting points of the primer along with primers in a table.

Response: A table containing sequence and melting points of all primers has been added to Supplementary Materials.

  1. In lung fibroblast isolation and culture, the viability of cells assed by ApoSENSORTMcell viability assay or trypan blue cell counting assay. Which method was used by the author in this study? 

Response: “Lung fibroblast isolation and culture” section was revised to clarify that trypan blue cell counting assay was used in this study.

  1. In collagen gel contraction assay, line no. 334, type 1 collagen from rat tail......., it may be mouse tail.

Response: Type I collagen from rat tail tendon is correct.

  1. In statistical analysis, which ANOVA was used? 

Response: Statistical analysis has been clarified to show that two-way ANOVA was used.

Round 2

Reviewer 1 Report

Comments and Suggestions for Authors

The authors have not adequately addressed the main point raised in this review.

It is needed to confirm the dysregulation of Iqgap1 in the bleomycin-treated mouse lung. This confirmation could be achieved by using the lung tissues through Western blotting, Real-time PCR, or immunostaining. Alternatively, assessing the expression of Iqgap1 in lung fibroblasts isolated from bleomycin-treated mice through Western blotting or Real-time PCR could be considered. These assessments should be conducted in wild-type mice rather than Iqgap1 knockout mice.

Round 3

Reviewer 1 Report

Comments and Suggestions for Authors

The authors failed to comprehend and address the comments from this review.

Previously, the authors identified consistently elevated levels of IQ motif-containing GTPase-activating protein (IQGAP1) in lung fibroblasts (LF) isolated from patients with scleroderma-associated interstitial lung disease (ILD). This study utilized a mouse animal model to further investigate. It was essential to confirm the dysregulation of Iqgap1 in the bleomycin-treated wild type mouse lung compared to PBS/saline-treated wild type mice lung.

In response, the authors demonstrated the absence of IQGAP1 in the lung tissue of IQGAP1 KO mice. This confirms that the KO mice have successfully eliminated the expression of Iqgap1. However, this does not definitively confirm whether bleomycin treatment affects Iqgap1 expression.

Author Response

Many thanks for your comment. In response to this, we have performed immunostainings with IQGAP1 antibody of lung tissues obtained from control and bleomycin-treated mice. Our data demonstrate significant increase of IQGAP1 in bleomycin-challenged mice as compared to control (saline-instilled) mice. These results were incorporated in Figure 2, as well as in Results and in Discussion sections (highlighted in red). Additionally, we have added 2 new citations (also highlighted in red).